# INFORMATION-PRESERVING REFORMULATION OF REASONING TRACES FOR ANTIDISTILLATION

## ABSTRACT

Recent advances in Large Language Models (LLMs) show that extending the length of reasoning chains significantly improves performance on complex tasks. While revealing these reasoning traces helps users better follow, verify, and learn from the model's problem-solving process, it also makes them highly vulnerable to unauthorized distillation. To mitigate this risk, proprietary model providers often adopt aggressive protection strategies, such as replacing detailed reasoning with brief summaries, which deprive users of valuable intermediate information. To address this trade-off, we propose PART, an information-preserving antidistillation reformulation of reasoning traces. Motivated by the difference between how humans understand reasoning traces and how LLMs exploit them for supervised fine-tuning, we design a simple but effective two-step reformulation: removing self-talk behaviors and reordering sub-conclusions. A small auxiliary model is trained to perform this reformulation, incurring minimal computational overhead. Extensive experiments demonstrate that PART consistently disrupts distillation across student models of different sizes and types on various reasoning benchmarks. For instance, when training on reformulated traces, even the performance of a large 32B student model decreases from 54.17 to 46.88 on AIME 2024, corresponding to a 13.5% degradation.

Figure 1: Overview of PART. Directly exposing original reasoning traces leaves them vulnerable to unauthorized distillation, whereas providing only summaries deprives users of the information contained in the reasoning process. PART introduces an information-preserving antidistillation approach through reformulation at both the token level and the structural level.

# 1 INTRODUCTION

Large language models (LLMs) have recently achieved remarkable progress in domains such as mathematics and programming, largely driven by the use of long reasoning traces under test-time scaling OpenAI (2024b); DeepSeek-AI et al. (2025). Beyond enhancing performance, these reasoning traces also allow users to gain insights into the model's problem-solving process, thereby improving interpretability and trustworthiness of LLMs' response. However, exposing original reasoning traces makes them highly vulnerable to unauthorized distillation. It has been shown that supervised fine-tuning (SFT) on as few as ten of thousands of reasoning traces suffices for student models to attain comparable reasoning capabilities, leading to intellectual property leakage Huang et al. (2024).

To mitigate this risk, existing proprietary models providers often adopt restrictive strategies to protect their reasoning traces. Common practices include either eliminating access to the reasoning trace or only revealing a condensed summary. While such strategies could prevent distillation, they hinder users from obtaining valuable information in reasoning traces. Recent works have explored antidistillation by controlling the sampling process or fine-tuning the teacher model Savani et al. (2025); Li et al. (2025c). However, these approaches either compromise the performance of the teacher model or incur training costs for large teacher models.

To address this issue, we introduce PART, a method that defend distillation by rewriting reasoning traces while preserving their information for human readers. The key insight of PART is that the way models acquire reasoning ability through SFT differs from how humans comprehend reasoning processes. Reasoning traces that are interpretable for humans may not be suitable for distillation Chen et al. (2025). Leveraging this discrepancy, we could defend distillation while preserving information. Concretely, we reformulate reasoning traces in two steps, modifying them at both the token level and the structural level, and we further train a small auxiliary model to perform this reformulation with minimal computational overhead.

At the token level, different tokens contribute unequally to parameter updates during SFT: tokens with lower predicted probabilities induce large gradients. Our analysis of student models' learning dynamics on teacher-generated sequences reveals that these low-probability tokens contain many self-talk behaviors such as "Hmm," "Wait," and "Let's." While such expressions do not carry reasoning-related information, they drive substantial loss reduction during training, acting as the "useful tokens" discussed in prior studies Lin et al. (2024). Removing these expressions therefore disrupt distillation without sacrificing informational content.

At the structural level, prior work has shown that the structural perturbations to reasoning traces can significantly impact distillation Li et al. (2025a). To construct an information-preserving structural perturbation, we exploit the difference between reasoning understanding and generation. Humans do not require strict process-then-conclusion order to comprehend reasoning. Instead, it is common to use conclusion-before-process structures, such as presenting a lemma before its proof in mathematics. In contrast, limited by single-step computation, it's hard for LLMs to directly generate the correct conclusion without intermediate reasoning steps. Based on this difference, we reorder reasoning traces by placing sub-conclusions ahead of their corresponding reasoning steps. This reordering perturbs the structural patterns on which distillation relies, thereby weakening its effectiveness while maintaining human interpretability.

To verify that PART effectively preserves information after reformulation, we evaluate the reformulated reasoning traces produced against the original reasoning traces from three perspectives: lexical similarity, semantic similarity, and human judgment. For lexical similarity, we segment the original reasoning traces into fragments and compute the match ratio in the reformulated reasoning traces using fuzzy matching. Experimental results show that across all similarity thresholds, PART consistently outperforms the summary-based method. For semantic similarity, we employed Qwen3-Embedding-4B Zhang et al. (2025) to map reasoning traces into embeddings and compute the match ratio by using the embeddings of the original traces as queries. Results demonstrate that 90.1% of queries matched the reformulated reasoning traces generated by PART, while only 7.3% matched those produced by the summary-based method. Furthermore, in a user study on perceived informativeness, participants generally judged the information in PART reformulations to be comparable to the originals, while clearly preferring PART over the summary-based method for providing richer information.

To evaluate the antidistillation capability of PART, we conducted experiments on student models of different sizes and types, comparing their performance when distilled with the original reasoning traces versus the reformulated reasoning traces. Results show that models trained on data reformulated by PART suffer significant degradation across mathematics, coding, and scientific question answering benchmarks. PART demonstrates stable effectiveness across varying model sizes and dataset scale. Notably, even a 32B student model exhibited a performance drop from 54.17 to 46.8 on AIME 2024, corresponding to a 13.5% degradation.

The contributions of this paper are summarized as follows:

- We propose PART, a simple but effective reasoning trace reformulation method that disrupts distillation while preserving information. We validate that our approach successfully retains the information contained in the original reasoning traces from multiple perspectives, including lexical similarity, semantic similarity, and human judgment.

- We conduct extensive distillation experiments and demonstrate that our method effectively degrades the performance of distilled models across student models up to 32B parameters, varying amounts of training data, and diverse downstream tasks.

- We will release the code and data to facilitate future research on antidistillation and reasoning trace reformulation.

## 2 METHOD

### 2.1 PROBLEM FORMULATION

Knowledge distillation aims to leverage a strong teacher model $T$ to guide a lightweight student model $S$, transferring the teacher's capabilities to the student Gou et al. (2021); Xu et al. (2024). A common approach to distilling large language models is supervised fine-tuning (SFT) on the data generated by the teacher. The student model is optimized to maximize the log-likelihood of the teacher's output $y$ conditioyned on query $q$:

$$\mathcal{L}_{\text{SFT}}(\theta_S) = -\frac{1}{T} \sum_{t=1}^{T} \log p_{\theta_S}(y_t \mid y_{<t}, q) \tag{1}$$

For reasoning models, each output sequence $y = (r, a)$ consists of a reasoning trace $r$ and a final answer $a$. To interfere with distillation, we consider a transformation $\mathcal{T} : r \mapsto r'$ that rewrites the reasoning trace. We keep the final answer unchanged, because it conveys the task outcome from which users extract the final result. Existing proprietary models often adopt restrictive disclosure strategies to prevent unauthorized distillation by others. For example, they omit the reasoning trace entirely or present only a high-level summary. However, these ways cause substantial information loss for users.

Our goal is therefore to design a transformation $\mathcal{T}$ that meets the following two objectives:

- Interfere with distillation. Make the distilled model $S_{\mathcal{D}_\mathcal{T}}$, which is trained on the modified dataset $\mathcal{D}_\mathcal{T} = \{(q_i, \mathcal{T}(r_i), a_i)\}_{i=1}^N$, yield degraded downstream performance $\text{Perf}(S_{\mathcal{D}_\mathcal{T}})$.

- Information Preservation. Ensure that the modified reasoning trace $\mathcal{T}(r_i)$ remains human-readable and preserves as much useful information as possible in each $r_i$, so that it stays interpretable and useful for human readers.

Formally, this trade-off can be posed as the constrained optimization problem:

$$\begin{aligned} \arg\min_{\mathcal{T}} \quad & \text{Perf}(S_{\mathcal{D}_\mathcal{T}}) \\ \text{s.t.} \quad & \text{Sim}(r_i, \mathcal{T}(r_i)) > \tau, \quad \forall i \end{aligned} \tag{2}$$

where $\text{Sim}(r_i, \mathcal{T}(r_i))$ is a similarity measure between the original and rewritten reasoning trace, and $\tau$ is a similarity threshold ensuring that $r'$ remains sufficiently faithful to $r$ from a human reader's perspective.

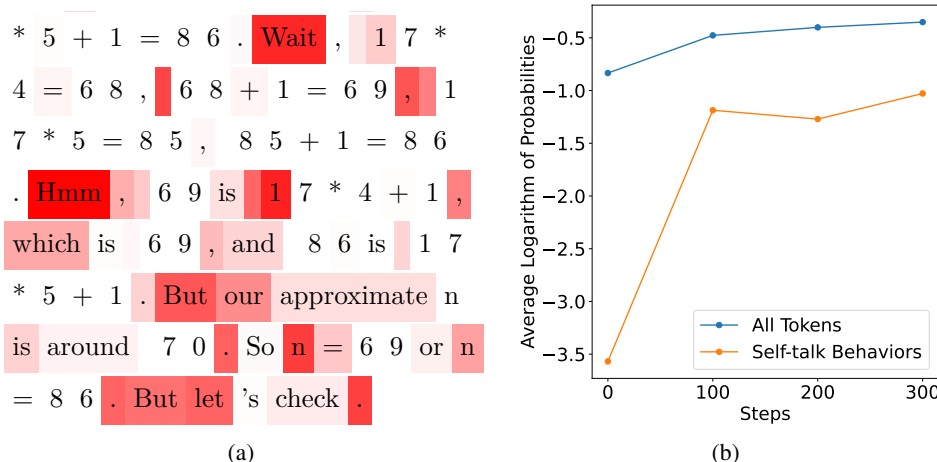

(a)                                     (b)

Figure 2: Predicted probabilities of the student model on teacher-generated reasoning traces. (a) Visualization of token-level predicted probabilities, where deeper red indicates lower probabilities. Teacher-generated traces exhibit frequent self-talk behaviors, which conveys little reasoning content yet receives low probabilities. (b) Tracking the probabilities of self-talk-behavior tokens across training stages reveals that they remain persistently lower than the average probabilities, suggesting that these semantically uninformative expressions exert disproportionate influence on gradient updates.

## 2.2 REASONING TRACE REFORMULATION

To construct an information-preserving antidistillation reformulation method, the key lies in identifying the differences between how LLMs learn reasoning through SFT and how humans comprehend reasoning traces. Prior studies have shown that reasoning traces that are easily understood by humans are not necessarily suitable for distillation; in fact, manually annotated chain-of-thought data sometimes perform poorly in distillation Chen et al. (2025). Leveraging this discrepancy, we design a reformulation method from two complementary perspectives: the token level and the structural level.

**1. Removing self-talk behaviors**

At the token level, we first analyze how different tokens contribute to the SFT from the gradient perspective. For a single time step $t$, $y_t$ denotes the ground-truth token at position $t$, and define the logits vector is denoted as $z^{(t)} \in \mathbb{R}^V$ over the vocabulary of size $V$. The token-level loss is

$$\mathcal{L}^{(t)}(\theta) = -\log p_{y_t}^{(t)} = -\log\left(\frac{e^{z_i^{(t)}}}{\sum_{j=1}^{V} e^{z_j^{(t)}}}\right). \tag{3}$$

The gradient of this loss with respect to the logits is

$$\nabla_{z^{(t)}} \mathcal{L}^{(t)} = p^{(t)} - e_{y_t}, \tag{4}$$

where $e_{y_t}$ is the one-hot indicator vector for the target token.

The squared $\ell_2$-norm of the gradient vector is

$$\left\|\nabla_{z^{(t)}} \mathcal{L}^{(t)}\right\|_2^2 = \sum_{i=1}^{V} \left(p_i^{(t)} - e_{y_t,i}\right)^2 = \sum_{i=1}^{V} (p_i^{(t)})^2 + 1 - 2p_{y_t}^{(t)} \tag{5}$$

This expression reveals a direct dependence on the predicted probability of the correct token $p_{y_t}^{(t)}$. When $p_{y_t}^{(t)} \to 1$, the gradient approaches zero. When $p_{y_t}^{(t)}$ is small, the gradient norm grows and signals a strong update. This indicates that tokens that the model already predicts with high confidence contribute negligible gradients and quickly fade from influencing optimization. In contrast,

low-probability (i.e., poorly predicted) tokens dominate the effective training signal, guiding the model to adjust its parameters toward correcting these mistakes.

Based on the analysis, we examine the predicted probabilities of the student model on teacher-generated reasoning traces. Figure 2(a) visualizes the predicted probabilities on a segment of a teacher-generated trace, where deeper red indicates lower probabilities. We observe that low-probability tokens contain many self-talk behaviors, a phenomenon where the model often speaks in the first person and employs colloquial expressions such as "hmm" and "wait". These expressions contain little information relevant to reasoning. However, the student model assigns low probabilities to such tokens, which results in large gradient.

We further track the predicted probabilities of representative tokens like "Hmm" and "Wait" across different training stages. As shown in Figure 2(b), these tokens exhibit persistently lower probabilities than the average token, which implies that such semantically uninformative expressions exert disproportionate influence during parameter updates. Previous studies have also explored the different influence of tokens during training. Lin et al. (2024) identified distinct loss patterns across tokens in pretraining: some tokens consistently maintain high or low loss, while only a subset exhibits significant loss reduction and are regarded as useful tokens. Tokens associated with self-talk behaviors demonstrate a similar pattern, indicating their impact on training.

To leverage this, we rewrite the reasoning traces to remove self-talk behaviors. This modification incurs negligible information loss, while deliberately perturbing the gradients associated with low-probability tokens, thereby affecting the distillation process.

**2. Reordering the sub-conclusions**

At the sequence level, LLMs learn to imitate the overall logical structure of reasoning traces in order to perform reasoning. Li et al. (2025a) demonstrates that structural perturbations to reasoning traces have a substantial impact on the performance of distilled models. However, their methods focused on operations such as randomly shuffling or deleting steps, or inserting irrelevant steps, which severely compromise human readability.

To design a form of structural perturbation that preserves readability for humans, we exploit the difference between generating a reasoning process and understanding a reasoning process. During reasoning, generation proceeds in a strictly sequential manner: the model must first produce intermediate steps before reaching the conclusion. In contrast, comprehension does not require this order; humans often prefer presenting the conclusion first, followed by the supporting process. For example, in mathematical reasoning, lemmas are often stated prior to their proofs, and in academic writing, abstracts precede detailed methods and results. This conclusion-before-process structure can even enhance human understanding of reasoning.

For LLMs, since the computational capacity per step is bounded, LLMs without chain-of-thought can only solve problems of limited complexity Li et al. (2024). This limitation makes it difficult for an LLM to distill reasoning traces with conclusion-before-process order, which breaks the chain-of-thought structure and makes the model struggle to directly generate correct conclusions.

Leveraging this asymmetry, we rewrite reasoning traces by reordering them into a conclusion-before-process structure. Specifically, we prompt GPT-4o OpenAI (2024a) to reformulate reasoning traces in a chain-of-thought style, where sub-conclusions are first summarized and then placed before their associated reasoning steps.

## 2.3 TRAINING A COMPACT REFORMULATION MODEL

In practical applications, it is desirable to minimize the cost of reformulation so as to mitigating its impact on the inference service.. To this end, we train a compact model for reformulation. Each original reasoning trace is divided into multiple segments, which are reformulated individually and then concatenated back into a complete trace. We fine-tune Qwen2.5-1.5B-Instruct Qwen et al. (2025) using paired data consisting of original reasoning traces and their rewritten counterparts generated by GPT-4o. Compared with advanced reasoning models such as DeepSeek-R1, the reformulation model introduces less than 1% additional parameters and incurs only about 4% extra computational overhead.

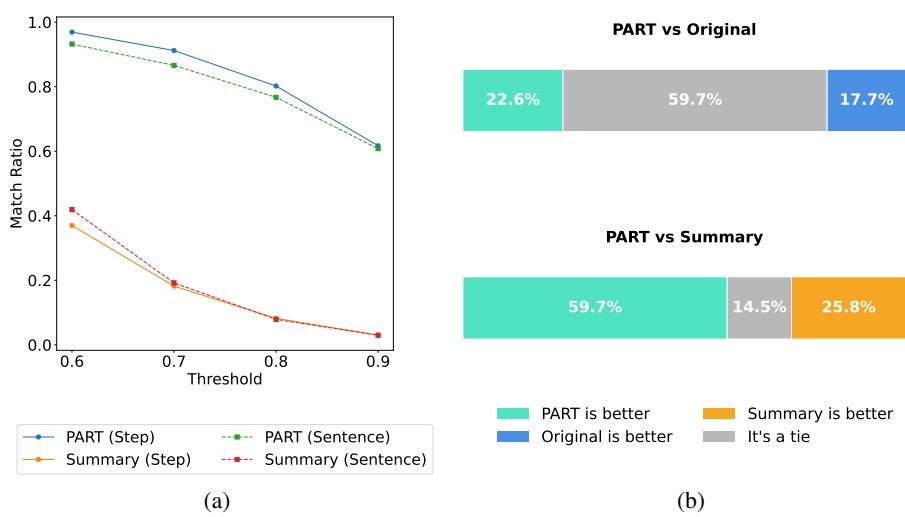

(a)                                                    (b)

Figure 3: (a) Match ratios under different lexical similarity score thresholds. PART achieves significant higher match ratios than the summary-based method at both step and sentence levels. (b) Human judgment about informativeness. Compared to original reasoning traces, PART is judged similarly informative; compared to the summary-based reformulation, PART is clearly preferred in terms of the informativeness.

Appendix E is an example of reformulation generated by our compact reformulation model. Despite its relatively small size, the model effectively accomplishes the reformulation task, successfully performing removal and reordering while preserving the information. In Section 4.3, we present a quantitative evaluation of the reformulation model, demonstrating its ability to achieve both antidistillation with information preservation.

# 3 QUALITY OF REFORMULATED REASONING TRACES

When reformulating reasoning traces to defend distillation, it is necessary not only to ensure the impact on the performance of distilled models but also to maintain the quality of the reformulated traces. In extreme cases, completely nonsensical reasoning traces would indeed prevent successful distillation, but they would also be unreadable to users and thus fail to convey any useful information. To this end, we evaluate the quality of reformulated reasoning traces using three complementary approaches: lexical similarity, semantic similarity, and human judgment. We compare the quality of traces reformulated by PART with that of the original traces and segment-level summaries. These methods verify that our reformulations can disrupt distillation while still preserving the essential information of the original reasoning traces, thereby ensuring usability for users.

## 3.1 LEXICAL SIMILARITY

A straightforward way to compare the lexical similarity between the original and reformulated reasoning traces is to perform fuzzy matching at the segment level. Specifically, we split the original reasoning trace into sentences or steps and check whether they can be successfully matched within the reformulated reasoning trace. We employ the `partial_ratio_alignment` function from the `RapidFuzz` library to calculate the similarity score, which first performs substring matching and then computes the normalized Indel similarity based on edit distance.

As shown in Figure 3(a), we compute the match ratio under different similarity score thresholds. Whether the matching is conducted at the step or sentence level, PART exhibits a remarkably high match ratio, substantially surpassing summary-based approaches across different thresholds. Examples of matched text pairs under different thresholds are shown in Appendix F. At a threshold of 0.7,

PART achieved a match ratio of 91%, whereas the summary-based method achieved only 18%. This indicates that PART is able to interfere with distillation through only minimal textual modifications.

## 3.2 SEMANTIC SIMILARITY

To compare the semantic similarity of reasoning traces, we employed Qwen3-Embedding-4B Zhang et al. (2025) to map reasoning traces into text embedding, under the assumption that semantically closer reasoning traces should yield higher embedding similarity. We treated the original reasoning traces as queries, and the reformulated reasoning traces produced by PART and the summary-based approach as candidate documents.

Experimental results show that 97.4% of queries successfully matched their corresponding reformulated reasoning traces. Specifically, 90.1% of queries matched the reasoning traces reformulated by PART, whereas only 7.3% matched those reformulated by the summary-based method. Moreover, the average cosine similarity between the original reasoning traces and those reformulated by PART reached 0.950, compared to 0.889 for the summary-based method. These results demonstrate that PART achieves superior semantic similarity to the original reasoning traces.

## 3.3 HUMAN JUDGMENT

To assess user perceptions of reformulated reasoning traces, we conducted a questionnaire study on their perceived informativeness. We sampled 50 original reasoning traces, each paired with two reformulated versions: one generated by PART and one by a summary-based method. Each participant evaluated four pairs of traces, indicating their preference ("A is better," "B is better," or "Tie"). Two pairs compared original traces with those reformulated by PART, and the other two compared summary-based reformulations with those from PART.

We collected 31 completed questionnaires, from which we obtained 124 comparisons. As shown in Figure 3(b), when comparing reasoning traces produced by PART with the original traces, most participants judged the information content to be comparable. In contrast, when comparing PART with the summary-based method, participants showed a clear preference for PART in terms of the richness of information provided.

## 4 EXPERIMENTS

### 4.1 SETUP

To assess the impact of our reasoning trace reformulation method on the effectiveness of distillation, we distill student models using both the original reasoning traces and the rewritten reasoning traces, and compare their performance.

**Training Setup.** We experiment with student models of different sizes and families. Specifically, we follow DeepSeek-R1 DeepSeek-AI et al. (2025) in selecting the base models: Qwen2.5-Math-1.5B, Qwen2.5-Math-7B Yang et al. (2024), Qwen2.5-14B, Qwen2.5-32B Qwen et al. (2025). In addition, we also examine distillation with an instruct model as the student model, for which we use Qwen2.5-7B-Instruct. Since the Qwen2.5-Math models only support a maximum context length of 4K tokens, we extend their context window by setting the `rope_theta` parameter to 1,000,000 following Liu et al. (2025). For distillation data, we use the Bespoke-Stratos-17k Labs (2025) and OpenThoughts-114k datasets Guha et al. (2025). We adopt the Llama-Factory Zheng et al. (2024) framework to perform SFT.

**Evaluation Setup.** For evaluation, we evaluate the distilled models on MATH-500 Hendrycks et al. (2021); Lightman et al. (2023), AIME 2024 MAA (2024), LiveCodeBench v2 Jain et al. (2024), and GPQA-Diamond Rein et al. (2023), covering tasks in mathematical reasoning, code generation, and scientific question answering. To obtain more reliable estimates of pass@1 accuracy, we sample multiple responses per query, thereby reducing variance in the results.

Table 1: Performance of distilled models on various benchmarks. "MATH500" refers to MATH-500, "AIME24" to AIME 2024, "LCBv2" to LiveCodeBench v2, and "GPQA-D" to GPQA-Diamond. More negative values of $\Delta$ indicate stronger antidistillation effects.

| Student Model | Data | MATH500 | AIME24 | LCBv2 | GPQA-D | Average |
|---|---|---|---|---|---|---|
| **Training Data: Bespoke-Stratos-17k** | | | | | | |
| Qwen2.5-Math-1.5B | original | 72.55 | 15.00 | 12.23 | 29.80 | 32.40 |
| | PART | 59.05 | 8.75 | 9.88 | 25.88 | 25.89 |
| | $\Delta$ | -13.50 | -6.25 | -2.35 | -3.92 | -6.51 |
| Qwen2.5-Math-7B | original | 88.95 | 32.71 | 34.88 | 43.18 | 49.93 |
| | PART | 80.00 | 22.08 | 28.96 | 38.00 | 42.26 |
| | $\Delta$ | -8.95 | -10.63 | -5.92 | -5.18 | -7.67 |
| Qwen2.5-14B | original | 90.60 | 43.75 | 55.58 | 53.28 | 60.80 |
| | PART | 82.25 | 25.83 | 43.98 | 46.97 | 49.76 |
| | $\Delta$ | -8.35 | -17.92 | -11.60 | -6.31 | -11.05 |
| Qwen2.5-32B | original | 92.65 | 54.17 | 70.99 | 61.24 | 69.76 |
| | PART | 89.65 | 46.88 | 62.38 | 55.68 | 63.65 |
| | $\Delta$ | -3.00 | -7.29 | -8.61 | -5.56 | -6.12 |
| Qwen2.5-7B-Instruct | original | 83.05 | 21.04 | 36.64 | 43.31 | 46.01 |
| | PART | 70.85 | 12.29 | 27.84 | 32.32 | 35.83 |
| | $\Delta$ | -12.20 | -8.75 | -8.80 | -10.99 | -10.19 |
| **Training Data: OpenThoughts-114k** | | | | | | |
| Qwen2.5-Math-7B | original | 90.40 | 46.67 | 41.98 | 45.20 | 56.06 |
| | PART | 78.50 | 28.54 | 30.87 | 36.49 | 43.60 |
| | $\Delta$ | -11.90 | -18.13 | -11.11 | -8.71 | -12.46 |

## 4.2 RESULTS

Table 1 reports the performance of distilled models across different benchmarks. It shows that, regardless of student model size or benchmark, PART consistently leads to a significant degradation in the performance of distilled models, thereby providing an effective defense against distillation. For example, even the performance of a large 32B student model decreases from 54.17 to 46.88 on AIME 2024, corresponding to a 13.5% degradation.

For the student model in reasoning distillation, some studies adopt a base model DeepSeek-AI et al. (2025); Liu et al. (2025), while others use an instruct model Labs (2025). We conduct experiments on both choices of student models. When using Qwen2.5-Math-7B as the student model, the average score decreases by 7.67. When using Qwen2.5-7B-Instruct as the student model, the score decreases by 10.19. This demonstrates that PART is effective against different types of student models.

## 4.3 EFFECTIVENESS OF THE REFORMULATION MODEL

To evaluate the generalization capability of the reformulation model, We trained a 1.5B reformulation model on reformulated data generated by GPT-4o using the Bespoke-Stratos-17k dataset and applied it to reformulate OpenThoughts-114k. As shown in Table 1, the reasoning traces produced by this small reformulation model also effectively defend against distillation: student models distilled on the reformulated data exhibit significant performance degradation.

We further evaluated the quality of traces generated by the reformulation model. For lexical similarity, the match ratio reached 88% under a threshold of 0.7. For semantic similarity, the average cosine similarity was 0.94. These similarity metrics are close to those obtained with GPT-4o reformulations and substantially higher than those of the summary-based method. This demonstrates that our reformulation model is also effective in preserving information during reformulation.

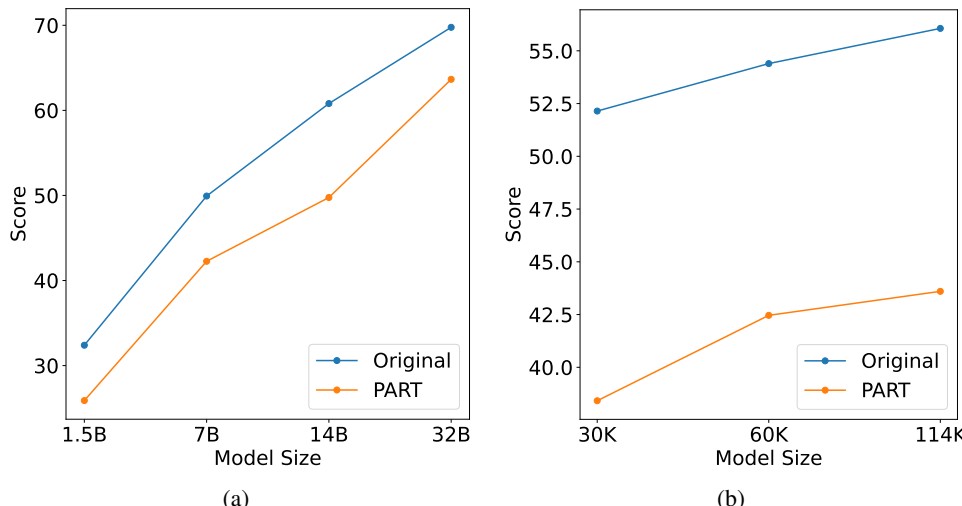

(a)                                                      (b)

Figure 4: Performance comparison (a) across different student model sizes and (b) across different data scales of distilled models trained on original versus reformulated traces. Across both factors, PART leads to consistent performance degradation of the distilled models, demonstrating its effectiveness as an antidistillation approach.

### 4.4 DETECTABILITY

An additional property of PART is detectability. Similar to LLM watermarking, models distilled on PART-reformulated data can be readily distinguished from those trained on original reasoning traces. Due to the removal of self-talk behaviors, the distributional patterns of the data undergo significant changes. We computed the term frequency of keywords related to self-talk behaviors: in the original traces, the average frequency of such keywords reached 2.9%, whereas in the reformulated traces it dropped to only 0.4%. Leveraging this substantial discrepancy, even a simple classifier based on a term-frequency threshold is sufficient to achieve separation, yielding an F1 score of 0.93 and a true-positive rate of 88.3% at a 1% false-positive rate (TPR@FPR).

### 4.5 ROBUSTNESS TO DATA SCALE

To evaluate the effectiveness of PART under varying amounts of training data, we sampled subsets of different sizes from the OpenThoughts-114K dataset and its corresponding reformulated traces. As shown in Figure 4(b), PART consistently led to a significant degradation in distilled model performance across different data scales. Notably, models trained on a large number of reformulated traces still underperformed compared to those trained on only a smaller number of original traces. This finding indicates that it is costly to collect more data to offset the impact of PART.

## 5 CONCLUSION

We presented PART, an information-preserving reformulation of reasoning traces for antidistillation. Leveraging the difference between how humans comprehend reasoning process and how LLMs acquire reasoning ability via supervised fine-tuning, PART applies two simple but effective steps: removing self-talk tokens and reorder sub-conclusions before their supporting process. Across lexical, semantic, and human-judgment evaluations, PART retains the information of original traces, substantially outperforming summary-based method. Distillation experiments show consistent degradation for student models trained on reformulated traces across various benchmarks, robust to model size and data scale. Overall, PART offers a practical method to balance interpretability with protection of model intellectual property.

## 6 ETHICS STATEMENT

This study involved the collection of human responses through questionnaires. All participants provided data voluntarily, and no personally identifiable or sensitive information was collected. The survey was designed to ensure anonymity and privacy, and all responses were analyzed in aggregate form only.

## 7 REPRODUCIBILITY STATEMENT

To help with reproducibility, we introduce our training and evaluation setup in Section 4.1 and hyperparameters in Appendix D. Prompts used in experiments are shown in Appendix C. We commit to release our code and data soon after the conference.

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

## A    THE USE OF LARGE LANGUAGE MODELS

In paper writing, we use Large Language Models to polish writing. In experiments, we use Large Language Models both as our research subjects and as tools to generate training data.

## B    RELATED WORK

**Reasoning Distillation**    With the success of test-time scaling OpenAI (2024b); DeepSeek-AI et al. (2025); Team (2025), an increasing number of studies have focused on distilling reasoning ability into smaller models. O1 Journey demonstrates that a base model fine-tuned on only tens of thousands of reasoning traces can outperform O1-preview Huang et al. (2024). DeepSeek-R1 adopts reinforcement learning for training, followed by distillation to obtain efficient smaller models DeepSeek-AI et al. (2025). In addition, several datasets have collected large-scale reasoning traces from advanced reasoning models—ranging from tens of thousands to millions—which have been used to train strong distilled models Labs (2025); Guha et al. (2025); Zhao et al. (2025).

**LLM Watermarking**    LLM watermarking focuses on tracking the text generated by LLMs and identifying whether a given piece of text was produced by a particular LLM. Common approaches achieve this by manipulating the sampling distribution during generation, ensuring detectability without compromising readability or fluency Kirchenbauer et al. (2023); Dathathri et al. (2024). Pan et al. (2025) further explores whether the watermark can still be detected when a student model is distilled using outputs from a protected LLM. While LLM watermarking is also related to model intellectual property detection, its primary emphasis is on post-hoc detection rather than proactively interfering with distillation.

**Unlearnable Data**    Unlearnable data focuses on perturbing training data to degrade model performance Li et al. (2025b). Li & Liu (2023) introduces hints into the input text, such as inserting class-wise symbols. RegText treats low-frequency, task-representative tokens as spurious words and randomly inserts these spurious words into the text. These approaches emphasize modifications to the input data, inducing models to rely on shortcuts Java et al. (2025). However, they are unsuitable for antidistillation, since we cannot alter the prompts used by the attacker. Moreover, such methods are typically limited to classification tasks. In contrast, our approach modifies the model's generation and does not rely on task-specific designs.

**Antidistillation**    Recent work has also begun to explore antidistillation for reasoning models. Antidistillation Sampling poisons reasoning traces by modifying a model's next-token probability distribution during sampling Savani et al. (2025). This method requires two auxiliary models: a proxy student model, and a variant of the proxy model obtained by performing a single step of gradient ascent on the downstream loss. At each reasoning step, the difference between the logits of these two models is computed to form a perturbation vector. Another method DOGe defends against distillation by fine-tuning the teacher model itself, jointly minimizing the SFT loss while maximizing the KL divergence between the teacher model and the proxy student model Li et al. (2025c). Both approaches interfere with the teacher model—either by altering its sampling distribution or modifying its parameters. Moreover, their effectiveness has only been demonstrated on small student models ($< 4B$ parameters). By contrast, PART introduces reasoning traces reformulation that do not affect the teacher model's ability to generate correct answers, and has been validated as effective across student models up to 32B in scale.

## C PROMPTS

---

**Removing Self-talk Behaviors**

---

Rewrite the given text, which is a part of a complete reasoning process. Convert only the parts expressed in a self-talk style into a declarative format. Avoid using first-person expressions such as 'I', 'me', 'we', or 'let's'. Do not alter any parts that are not self-talk; keep them exactly as in the original text.
Do not add any extra information. Do not include any introductory phrases.
Text:

---

**Reordering the Sub-conclusions**

---

You will process the given text in two steps. The given text is a part of a complete reasoning process.
Step 1: Extract and list the most important sub-conclusions in the given reasoning process. Keep the number of sub-conclusions small and focused.
Wrap the sub-conclusions in the tags `` and `` for easy extraction.
Step 2: Move the sentences corresponding to these sub-conclusions to appear *before* their respective reasoning processes. Keep the sub-conclusions unnumbered and naturally integrated into the context. Do not modify any other parts of the original text.
Wrap the entire transformed text in the tags `<REWRITTEN>` and `</REWRITTEN>` for easy extraction.
Text:

## D HYPERPARAMETER

Table 2: Hyperparameters used in training. We use larger learning rate for smaller model.

| Hyperparameter | Default | 1.5B Model |
|---|---|---|
| Cutoff length | | 16,384 |
| Batch size | | 128 |
| Learning rate | 3.0e-5 | 1.0e-4 |
| Epochs | | 3 |
| LR scheduler | | Cosine |
| Warmup ratio | | 0.1 |

Table 3: Hyperparameters used in evaluation. For AIME24, we increase the number of responses per query due to the small number of problems.

| Hyperparameter | Default | AIME24 |
|---|---|---|
| Temperature | | 0.6 |
| Top-$p$ | | 0.95 |
| Maximum number of tokens to generate | | 32,768 |
| Number of responses per query | 4 | 16 |

# E   EXAMPLE OF REFORMULATED REASONING TRACE

## Original Traces

Okay, let's see. I need to solve this problem where three people (Chef, head server, sous-chef) have transceivers that can communicate directly if within R meters. If not, but there's a third person acting as a bridge, then they can still communicate. The goal is to check if all three can communicate with each other, possibly through the third.

Hmm. So for each test case, given R and three points, determine if the three form a connected graph where each pair is either directly connected (distance $\leq$ R) or connected via the third.

Wait, but how exactly? Let's think. The communication can go through one intermediary. So all three must be in a chain where each consecutive pair is within R, or perhaps any two are connected via a path through the third.

## Reformulated Traces

The goal is to check if all three can communicate with each other, possibly through the third. The problem involves three people (Chef, head server, sous-chef) who have transceivers that can communicate directly if within R meters. If not, but there's a third person acting as a bridge, then they can still communicate.

For each test case, given R and three points, it is necessary to determine if the three form a connected graph where each pair is either directly connected (distance $\leq$ R) or connected via the third.

The condition is that all three can communicate with each other, possibly through one another. The communication can go through one intermediary. Therefore, all three must be in a chain where each consecutive pair is within R, or any two are connected via a path through the third.

# F   LEXICAL SIMILARITY

## Similarity: 0.63

**Original Step:**
But wait, let me verify this again to be sure. Because sometimes with clock angle problems, there can be another instance where the angle is 110 degrees, but depending on the direction (whether the minute hand is ahead or behind the hour hand). Let me think.

**Matched Part of Reformulated Trace:**
Verification is necessary to ensure accuracy, as there can be another instance where the angle is 110 degrees, depending on the direction of the hands. The formula $|30H - 5.5M|$ provides the absolute angle between the two hands.

## Similarity: 0.75

**Original Step:**
Okay, let me try to figure out this problem. So, we have a set of consecutive positive integers starting from 1, and one number is erased. The average of the remaining numbers is 35 and 7/17. We need to find out which number was erased. The options are from 6 to 9, or cannot be determined. Hmm.

**Matched Part of Reformulated Trace:**
The problem involves a set of consecutive positive integers starting from 1, with one number erased. The average of the remaining numbers is 35 and 7/17. The objective is to identify the erased number. The options range from 6 to 9, or indicate that the number cannot be determined.

**Similarity: 0.81**

**Original Step:**
Wait, just to be thorough, let's make sure none of the other options could work. Let's check option E: y = 10ˆx. If we take x = 0, then y = 1, which would correspond to the rotated point (0,1), which is correct. But let's take another point. If x = -1, y = 10ˆ(-1) = 0.1, but according to G', when x = -1, y should be 10. But according to option E, y = 10ˆ(-1) = 0.1, which is wrong. So E is out. Option D, as we saw, gives y = 10ˆ1 = 10 when x = -1, which is correct.

**Matched Part of Reformulated Trace:**
Check option E: y = 10ˆx. Taking x = 0, then y = 1, which would correspond to the rotated point (0,1), which is correct. However, taking another point: if x = -1, y = 10ˆ(-1) = 0.1, but according to G', when x = -1, y should be 10. According to option E, y = 10ˆ(-1) = 0.1, which is wrong. Thus, E is out. Option D gives y = 10ˆ1 = 10 when x = -1, which is correct.

**Similarity: 0.96**

**Original Step:**
So Thuy is 21, Kareem 22. So Kareem is indeed higher than Thuy, and Jose is the lowest of the three. Wait, but Thuy is 21, Jose is 20, Kareem 22. So the order from highest to lowest is Kareem, Thuy, Jose. So the largest is Kareem. Therefore, answer C.

**Matched Part of Reformulated Trace:**
So Thuy is 21, Kareem 22. So Kareem is indeed higher than Thuy, and Jose is the lowest of the three. Thuy is 21, Jose is 20, Kareem 22. So the order from highest to lowest is Kareem, Thuy, Jose. So the largest is Kareem. Therefore, answer C.

