# OpenReview forum: "Information-Preserving Reformulation of Reasoning Traces for Antidistillation"
_ICLR.cc/2026/Conference — ICLR 2026 Conference Withdrawn Submission_

### Official Review · Reviewer_PMVj · 2025-10-30

**Soundness:** 2
**Presentation:** 3
**Contribution:** 1
**Rating:** 2
**Confidence:** 4

**Summary:**

This paper focused on preventing unauthorized distillation behaviors, named “anti-distillation”, which is a rare but interesting topic. This work proposed an information-preserving anti-distillation reformulation method, named PART, to protect proprietary reasoning traces, which can mitigate unauthorized distillation while preserving enough information to understand. Empirical results illustrated PART's effectiveness during student LLMs distilling on reformulated reasoning traces.

**Strengths:**

1. This paper focused on an important and timely problem for intellectual property protection, especially in the era of distillation-enhanced reasoning LLMs.

2. By modifying the original reasoning paths while not affecting the main information, the proposed PART can mitigate unauthorized distillation and preserving the readability of the original reasoning LLMs as demonstrated in the experiments.

**Weaknesses:**

1. The anti-distillation mechanism is mainly for closed-source reasoning LLMs, such as some OpenAI models and some Anthropic models, which have developed strong defense for distillation. As open-source LLMs were made publicly available by their developers, the distillation-based reasoning ability enhancement is very insightful for the open-source community. From both the strong anti-distillation results of closed LLMs and distillation-free open-source LLMs, I don’t think the anti-distillation task is that contributing.

2. All analysis and validation experiments were conducted on DeepSeek-R1-like reasoning LLMs (given the training data are OpenThoughts-114k and Bespoke-Stratos-17k), as the R1-style is specific and not general, making all the findings and empirical results limited to one single reasoning style. Concretely, the advanced DeepSeek-R1-0528 has a different reasoning style, which is less self-talk words (e.g., “Hmm”, “wait”, “alternatively”), not to mention the open-source GPT-oss models whose reasoning styles are totally different from the DeepSeek-R1 series.

3. There is no consideration of attacks to recover the original reasoning paths. What if an authorized developer analyzed the revised reasoning cases, given they are still traceable from human perspective, then designed a heuristic or prompt-based method to complement the revised reasoning path to distillation-suitable path? There is no discussion on similar circumstances.

4. There have been already some anti-distillation defense approaches, such as injecting noise, and embedding watermarks [1,2]. However, this paper didn’t compare the proposed PART against these baselines to demonstrate its superiority. Without such a comparison, it is hard to determine whether PART could advance the protection–utility trade-off or merely works in the specific experimental setting. Such an omission significantly undermines the novelty and practical contribution of this work.

5. There is no anonymous code link or supplemental materials, making it hard to reproduce the results of this paper. Apart from that, the PART relies heavily on GPT-4o for reformulating training data, limiting the method’s generalizability.

[1] Instructional Fingerprinting of Large Language Models. NAACL 2024.

[2] Can LLM Watermarks Robustly Prevent Unauthorized Knowledge Distillation? ACL 2025.

[3] DOGe: Defensive Output Generation for LLM Protection Against Knowledge Distillation. arXiv: 2505.19504.

**Questions:**

1. How does the PART perform on reasoning traces that naturally lack self-talk behaviors, such as the GPT-oss models’ reasoning traces?

2. What are the extra computational costs/budgets of the reformulation process via calling another strong LLM? And how to guarantee the quality of the reformulation reasoning data?

3. Could the removal of self-talk behaviors actually improve distillation in some cases by removing noise, contrary to the paper's claims?

4. How sensitive are the results to different reordering strategies? Would different reordering approaches produce similar effects?

5. The reformulation requires to employ another model, have the authors considered that such an operation may also lead to reasoning data leakage when calling APIs?

---

### Official Review · Reviewer_focr · 2025-10-31

**Soundness:** 1
**Presentation:** 3
**Contribution:** 2
**Rating:** 2
**Confidence:** 3

**Summary:**

This submission is on the prevention of unauthorized distillation from (i.e., supervised fine-tuning on) the reasoning traces of large reasoning models. It proposes a method called PART, which aims to "reformulate" reasoning traces while preserving information by removing self-talk words and moving conclusions to occur before their corresponding reasoning steps. Reformulation is done using both GPT-4o and a fine-tuned Qwen2.5-1.5B-Instruct model. The quality of the reformulated reasoning traces is evaluated using lexical similarity, semantic similarity, and human preference. The reformulated traces are used to distill into student models of different sizes, showing consistent decreases in distilled model performance.

**Strengths:**

- PART uses simple, easy-to-understand operations
- I appreciate how the quality of the reformulated reasoning traces (i.e., their similarity to the original traces) was evaluated in three different ways, especially the human evaluation in Section 3.3.

**Weaknesses:**

1. A major issue for me is that it is unclear which results were obtained using GPT-4o for reformulation and which using the fine-tuned Qwen2.5-1.5B-Instruct model. Given the presence of Section 2.3, I initially thought that all results were obtained using the 1.5B model, including those on quality in Section 3. However, Section 4.3 suggests that most results are from GPT-4o instead, with only those mentioned in Section 4.3 from the 1.5B model. Stepping back a bit, I think that it would not be desirable to assume that a model like GPT-4o is always available for reformulation. Because of this, and also given the claims of "minimal computational overhead" in the abstract and introduction, I think that the paper should report all results using both models, i.e., from two instantiations of PART using GPT-4o or Qwen2.5-1.5B-Instruct.
    - A minor related comment is that it is not clear whether GPT-4o is also used to remove self-talk (although I would assume so).
1. I find it difficult to judge how significant are the degradations shown in Table 1. How much of the reasoning ability has been lost due to PART? One comparison that is missing in this regard is the performance of the student models before distillation.
1. Related to the previous point, Table 1 does not have results for other reformulation methods, not even the summarization method that was compared in Section 3. Moreover, lines 069-070 cite recent works on anti-distillation. While the drawbacks of these methods are mentioned, we do not know how well they perform in terms of degrading distillation.

Minor:
- Lines 072-073 and elsewhere: The phrase "defend(s) distillation" should be "defend(s) against distillation" since distillation is the thing being prevented, not itself defended.
- Lines 198-199, "define the logits vector is denoted as": Some extra words here.
- Eq. (3): In the numerator, I think the subscript should be $y_t$ instead of $i$.
- Can a statistical test be applied to assess the significance of the preference in Figure 3(b)?
- Appendix E: It would help to show correspondences between parts (e.g. steps) of the traces.

**Questions:**

1. Why do the authors use the term "intellectual property leakage" (in line 063 for example)? To my mind, the problem is about preventing others from taking advantage of the considerable effort that a model developer may have put into training an LLM for reasoning.
1. In lines 091-092, I am not sure what is meant by "limited by single-step computation." Similarly in line 251, please elaborate on "the computational capacity per step is bounded."
1. Line 113: Why is it notable that "even a 32B student model exhibited a performance drop"? Is it known that larger models are more adept learners and thus harder to disrupt?
1. Lines 265-266: What is the reason for dividing a reasoning trace into segments and reformulating individually?

---

### Official Review · Reviewer_97g5 · 2025-11-01

**Soundness:** 4
**Presentation:** 2
**Contribution:** 2
**Rating:** 4
**Confidence:** 2

**Summary:**

This paper presents PART, an information-preserving antidistillation method that reformulates reasoning traces to hinder unauthorized distillation while maintaining their usefulness for humans. It exploits a mismatch between how humans understand reasoning and how LLMs learn it via supervised fine-tuning. PART performs two complementary edits: at the token level, it removes self-talk fillers (e.g., “Hmm,” “Wait,” first-person phrases) that carry little reasoning content yet produce large gradients during SFT, thereby disrupting learning signals without losing information; at the structural level, it reorders traces to present sub-conclusions before the supporting steps, which aligns with human comprehension but breaks the chain-of-thought patterns students rely on to imitate reasoning.

**Strengths:**

(1) Clear, simple, and practical reformulation pipeline that does not alter final answers or require teacher model changes, enabling low-overhead deployment while preserving interpretability.

(2) Insightful gradient-based rationale for removing self-talk tokens; convincing evidence that such tokens carry low semantics but large gradients, hence disproportionately drive SFT.

(3) Well-motivated structure-level reordering (conclusion-before-process) that aligns with human comprehension yet perturbs student imitation of chain-of-thought generation.

(4) Strong, broad empirical validation: consistent degradation across tasks (math, code, science), student sizes up to 32B, and data scales, with stability across two public reasoning datasets.

(5) Information preservation is measured from multiple angles (lexical fuzzy matching, embedding similarity, human studies) and surpasses summary-only defenses; the compact reformulator generalizes.

**Weaknesses:**

(1) Adaptive attacker strategies are underexplored: e.g., filtering self-talk tokens, restoring process-then-conclusion order, using RLHF or structure-aware objectives, or data augmentation to undo PART.

(2) Structural operation focuses on moving sub-conclusions; other minimally invasive permutations (e.g., grouping, hierarchical chunking, proof sketches) and their trade-offs are not systematically compared.

(3) The token-level removal may also eliminate subtle metacognitive cues that sometimes aid human auditing; human evaluations cover informativeness, but not audit efficiency or error-detection ability.

(4) Reformulation effectiveness relies on an external generator (GPT-4o) for supervision; robustness to domain shifts (code, formal proofs, long-context) and to non-English corpora needs further evidence.

(5) Distillation setups are SFT-centric; results under more advanced student training (policy distillation, DPO/RLAIF, multi-step imitation, contrastive or step-supervision) are not reported.

(6) Detectability may enable adversaries to identify and counteract PART data during training; an analysis of stealth vs. robustness trade-offs is missing.

**Questions:**

(1) How robust is PART to adaptive distillers that (a) strip self-talk tokens, (b) reorder traces back to process-first, (c) train with step-level alignment or contrastive losses to recover structure, or (d) apply RLHF to penalize over-reordered traces?

(2) Can you quantify the impact of token removal on human audit tasks (error finding, step verification time) versus originals and summaries to ensure no loss in audit effectiveness?

(3) Beyond conclusion-first, which structural perturbations (e.g., lemma/fact grouping, proof sketches, section headers, hierarchical outlines) best preserve human utility while degrading distillation?

(4) How does PART perform on long-context reasoning, program synthesis with execution traces, formal proofs, and multilingual traces? Any failure modes where reordering harms human clarity?

(5) Against stronger distillation pipelines (e.g., step-supervised SFT, imitation of latent plans, DPO/RLAIF with preference data), does PART still induce comparable degradation?

(6) Can you add an adaptive reformulator that tunes removal/reordering intensity, for example, based on predicted student vulnerability or similarity constraints, and report accuracy–similarity trade-off curves?

---

### Official Review · Reviewer_a9HH · 2025-11-01

**Soundness:** 2
**Presentation:** 2
**Contribution:** 2
**Rating:** 2
**Confidence:** 3

**Summary:**

This paper proposes PART, a method to protect reasoning models from unauthorized distillation while preserving human interpretability. PART reformulates traces by: (1) removing self-talk tokens ("Hmm," "Wait") that receive low probabilities but drive large gradients, and (2) reordering sub-conclusions before their reasoning steps. A 1.5B model performs reformulation with 4% overhead. Results show consistent distillation degradation across 1.5B-32B students (e.g., 54.17→46.88 on AIME, -13.5%) while maintaining 91% lexical match, 90% semantic similarity, and positive human judgment versus summaries.

**Strengths:**

**Well-Motivated Problem**: Addresses the practical dilemma that exposing reasoning traces enables distillation while hiding them reduces interpretability. Clear formalization (Eq. 2) as constrained optimization: minimize distillation performance subject to information preservation.

**Principled Design**: Token-level intervention grounded in gradient analysis (Eq. 3-5) showing low-probability tokens dominate updates. Figure 2 validates self-talk tokens persistently receive lower probabilities. Structure-level design exploits generation-comprehension asymmetry: LLMs need sequential CoT to generate but humans understand conclusion-first order.

**Strong Information Preservation**: Three-pronged validation convincingly demonstrates preservation: 91% lexical match vs. 18% for summaries (0.7 threshold), 90.1% semantic match vs. 7.3%, and clear human preference (59.7% tie with originals, 59.7% vs. 14.5% over summaries).

**Weaknesses:**

**Missing Baselines and Comparisons**:

- Undefined "summary-based method"—unclear prompts/models used
- No simple baselines: random deletion, token shuffling, noise injection, differential privacy
- No cost-benefit analysis: latency, memory, protection-cost trade-offs

**Limited Mechanistic Understanding**: Gradient analysis explains *why* low-probability tokens matter but not *how* reformulation disrupts learning. Missing:  ablation isolating token vs. structure contributions, explanation for scale-dependent degradation (1.5B: -6.51%, 14B: -11.05%, 32B: -6.12%), theoretical bounds on expected degradation.

**Narrow Experimental Scope**:

- Single student family (Qwen2.5 only)—no Llama, Mistral, DeepSeek students
- Limited tasks (math/code/science)—missing creative, multi-modal, interactive reasoning
- No adversarial evaluation: mixing clean/reformulated data, denoising attacks, data augmentation, iterative refinement

**Questions:**

1. Lack of ablation:  Isolate token-level vs. structure-level vs. combined effects. What's the independent contribution of each?

2. Generalization: Test with different teachers (GPT-4o, Claude) and students (Llama, Mistral). Does it work on creative/multi-modal tasks?

3. Baseline comparison: Benchmark against Antidistillation Sampling, DOGe, random deletion. How does protection strength vs. cost compare? Which reasoning steps are most altered? Can students still learn from reformulated traces? Conduct expert evaluation.

---

### Note · Authors · 2026-01-05

I have read and agree with the venue's withdrawal policy on behalf of myself and my co-authors.